# Part-dependent Label Noise:
# Towards Instance-dependent Label Noise

**Xiaobo Xia**[1,2]    **Tongliang Liu**[1][†]  **Bo Han**[3]   **Nannan Wang**[2]
**Mingming Gong**[4]    **Haifeng Liu**[5]    **Gang Niu**[6]    **Dacheng Tao**[1]   **Masashi Sugiyama**[6,7]
[1]University of Sydney    [2]Xidian University
[3]Hong Kong Baptist University    [4]University of Melbourne
[5]Brain-Inspired Technology Co., Ltd    [6]RIKEN    [7]University of Tokyo

## Abstract

Learning with the *instance-dependent* label noise is challenging, because it is hard to model such real-world noise. Note that there are psychological and physiological evidences showing that we humans perceive instances by decomposing them into parts. Annotators are therefore more likely to annotate instances based on the parts rather than the whole instances, where a wrong mapping from parts to classes may cause the instance-dependent label noise. Motivated by this human cognition, in this paper, we approximate the instance-dependent label noise by exploiting *part-dependent* label noise. Specifically, since instances can be approximately reconstructed by a combination of parts, we approximate the instance-dependent *transition matrix* for an instance by a combination of the transition matrices for the parts of the instance. The transition matrices for parts can be learned by exploiting anchor points (i.e., data points that belong to a specific class almost surely). Empirical evaluations on synthetic and real-world datasets demonstrate our method is superior to the state-of-the-art approaches for learning from the instance-dependent label noise.

## 1   Introduction

Learning with noisy labels can be dated back to [4], which has recently drawn a lot of attention, especially from the deep learning community, e.g., [52, 79, 25, 13, 50, 57, 77, 37, 70, 75, 18, 41, 53, 22, 40, 56, 17, 16, 61, 60, 33, 32, 31, 21, 39, 46, 72, 66, 78]. The main reason is that it is expensive and sometimes even infeasible to accurately label large-scale datasets [23]; while it is relatively easy to obtain cheap but noisy datasets [77, 62, 65, 71, 19].

Methods for dealing with label noise can be divided into two categories: model-free and model-based algorithms. In the first category, many heuristics reduce the side-effects of label noise without modeling it, e.g., extracting *confident examples* with small losses [18, 75, 64]. Although these algorithms empirically work well, without modeling the label noise explicitly, their reliability cannot be guaranteed. For example, the small-loss-based methods rely on accurate *label noise rates*.

This inspires researchers to model and learn label noise [13, 54, 55]. The *transition matrix* $T(\boldsymbol{x})$ (i.e., a matrix-valued function) [44, 9] was proposed to explicitly model the generation process of label noise, where $T_{ij}(\boldsymbol{x}) = \Pr(\bar{Y} = j | Y = i, X = \boldsymbol{x})$, $\Pr(A)$ denotes as the probability of the event $A$, $X$ as the random variable for the instance, $\bar{Y}$ as the noisy label, and $Y$ as the latent clean label. Given the transition matrix, an optimal classifier defined by clean data can be learned by exploiting sufficient noisy data only [50, 35, 77]. The basic idea is that, the *clean class posterior* can be inferred by using the *noisy class posterior* (learned from the noisy data) and the transition matrix [6].

---

[†]Correspondence to Tongliang Liu (tongliang.liu@sydney.edu.au).

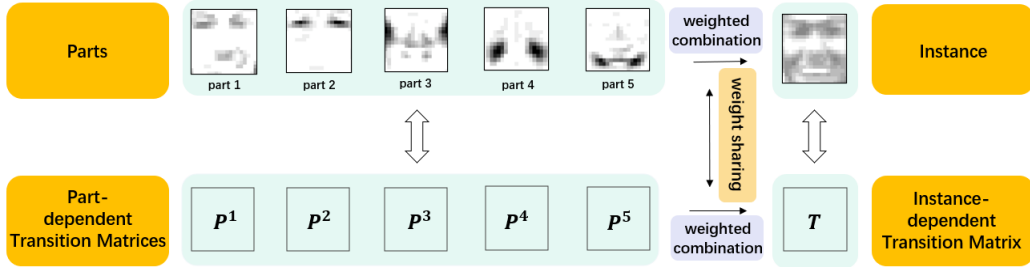

Figure 1: The proposed method will learn the transition matrices for parts of instances. The instance-dependent transition matrix for each instance can be approximated by a weighted combination of the part-dependent transition matrices.

However, in general, it is *ill-posed* to learn the transition matrix $T(\boldsymbol{x})$ by only exploiting noisy data [9, 67], i.e., the transition matrix is unidentifiable. Therefore, some assumptions are proposed to tackle this issue. For example, additional information is given [6]; the matrix is symmetric [43]; the noise rates for instances are upper bounded [9], or even to be instance-independent [67, 17, 50, 48, 44], i.e., $\Pr(\bar{Y} = j | Y = i, X = \boldsymbol{x}) = \Pr(\bar{Y} = j | Y = i)$. Note that there are specific applications where these assumptions are valid. That being said, in practice, these assumptions are hard to verify, and the gaps are large between instance-independent and instance-dependent transition matrices.

To handle the above problem, in this paper, we propose a new but practical assumption for instance-dependent label noise: *The noise of an instance depends only on its parts*. We term this kind of noise as *part-dependent* label noise. This assumption is motivated by that annotators usually annotate instances based on their parts rather than the whole instances. Specifically, there are psychological and physiological evidences showing that we humans perceive objects starting from their parts [49, 63, 38]. There are also computational theories and learning algorithms showing that object recognition rely on parts-based representations [7, 59, 10, 47, 20, 3]. Since instances can be well reconstructed by combinations of parts [29, 30], the part-dependence assumption should be mild in this sense. Intuitively, for a given instance, a combination of part-dependent transition matrices can well approximate the instance-dependent transition matrix, which will be empirically verified in Section 4.2.

To fulfil the approximation, we need to learn the transition matrices for parts and the combination parameters. Since the parts are semantic [29], their contributions to perceiving the instance could be similar in the contributions to understanding (or annotating) them [7, 3]. Therefore, it is natural to assume that for constructing the instance-dependent transition matrix, the combination parameters of part-dependent transition matrices are identical to those of parts for reconstructing an instance. We illustrate this in Figure 1, where the combinations in the top and bottom panels share the same parameters. The transition matrices for parts can be learned by exploiting *anchor points*, which are defined by instances that belong to a specific clean class with probability one [35]. Note that the assumption for combination parameters and the requirement of anchor points might be strong. If they are invalid, the part-dependent transition matrix might be poorly learned. To solve this issue, we also use the slack variable trick in [67] to modify the instance-dependent transition matrix.

Extensive experiments on both synthetic and real-world label-noise datasets show that the part-dependent transition matrices can well address instance-dependent label noise. Specifically, when the instance-dependent label noise is heavy, i.e., 50%, the proposed method outperforms state-of-the-art methods by almost 10% of test accuracy on *CIFAR-10*. More details can be found in Section 4.

The rest of the paper is organized as follows. In Section 2, we briefly review related work on modeling label noise and parts-based learning. In Section 3, we discuss how to learn part-dependent transition matrices. In Section 4, we provide empirical evaluations of our learning algorithm. In Section 5, we conclude our paper.

## 2 Related Work

**Label noise models** Currently, there are three typical label noise models, i.e., the random classification noise (RCN) model [8, 44, 42], the class-conditional label noise (CCN) model [50, 67, 79],

and the instance-dependent label noise (IDN) model [6, 9, 12]. Specifically, RCN assumes that clean labels flip randomly with a constant rate [1, 4, 24]; CCN assumes that the flip rate depends on the latent clean class [40, 18, 75]; IDN considers the most general case of label noise, where the flip rate depends on its instance. However, IDN is non-identifiable without any additional assumption, which is hard to learn with only noisy data [67]. The proposed part-dependent label noise (PDN) model assumes that the label noise depends on parts of instances, which could be an important "intermediate" model between CCN and IDN.

**Estimating the transition matrix**   The transition matrix bridges the class posterior probabilities for noisy and clean data. It is essential to build *classifier-/risk-consistent* estimators in label-noise learning [50, 35, 54, 77]. To estimate the transition matrix, a *cross-validation* method is used for the binary classification task [44]. For the multi-classification task, the transition matrix could be learned by exploiting anchor points [50, 76, 73]. To remove strong dependence on anchor points, data points having high noisy class posterior probabilities (similar to anchor points) can also be used to estimate the transition matrix via a slack variable trick [67]. The slack variable is added to revise the transition matrix, which can be learned and validated together by using noisy data.

**Parts-based learning**   Non-negative matrix factorization (NMF) [11] is the representative work of parts-based learning. It decomposes a non-negative data matrix into the product of two non-negative factor matrices. In contrast to principal components analysis (PCA) [2] and vector quantization (VQ) [14] that learn holistic but not parts-based representations, NMF allows additive but not subtractive combinations. Several variations extended the applicable range of NMF methods. For example, convex-NMF [34] restricts the basis vectors to be convex combinations of data. ONMF [74] imposes orthogonality constraints on data matrix, which achieves better performance than standard NMF in some applications. Semi-NMF [58] allows the data matrix and basis vectors to have mixed signs. LCNMF [36] pushes the simplicial cone spanned by the bases to be large, and thus makes the learning algorithms robust. Truncated CauchyNMF [15] can handle outliers by truncating large errors, which robustly learns the basis vectors on noisy datasets contaminated by outliers.

# 3   Part-dependent Label Noise

**Preliminaries**   Let $\bar{S} = \{(\boldsymbol{x}_i, \bar{y}_i)\}_{i=1}^n$ be the noisy training sample that contains instance-dependent label noise. Our aim is to learn a robust classifier from the noisy training sample that could assign clean labels for test data. In the rest of the paper, we use $A_{i\cdot}$ to denote the $i$-th row of the matrix $A$, $A_{\cdot j}$ the $j$-th column of the matrix $A$, and $A_{ij}$ the $ij$-th entry of the matrix $A$. We will use $\|\cdot\|_p$ as the $\ell_p$ norm of the matrices or vector, e.g., $\|A\|_p = \left(\sum_{ij} |A_{ij}|^p\right)^{1/p}$.

**Learning parts-based representations**   NMF has been widely employed to learn parts-based representations [11]. Many variants of NMF were proposed to enlarge its application fields [15, 36, 74], e.g., allowing the data matrix or/and the matrix of parts to have mixed signs [34]. For our problem, we do not require the matrix of parts to be non-negative, as our input data matrix is not restricted to be non-negative. However, we require the combination parameters (as known as new representation in the NMF community [11, 36, 15]) for each instance to be not only non-negative but also to have a unit $\ell_1$ norm. This is because we want to treat the parameters as the weights that measure how much the parts contribute to reconstructing the corresponding instance.

Let $\boldsymbol{X} = [\boldsymbol{x}_1, \ldots, \boldsymbol{x}_n] \in \mathbb{R}^{d \times n}$ be the data matrix, where $d$ is the dimension of data points. The parts-based representation learning for the part-dependent label noise problem can be formulated as

$$\min_{W \in \mathbb{R}^{d \times r}, \boldsymbol{h}(\boldsymbol{x}_i) \in \mathbb{R}_+^r, \|\boldsymbol{h}(\boldsymbol{x}_i)\|_1 = 1, i=1,\ldots,n} \quad \sum_{i=1}^n \|\boldsymbol{x}_i - W\boldsymbol{h}(\boldsymbol{x}_i)\|_2^2, \tag{1}$$

where $W$ is the matrix of parts (each column of $W$ denotes a part of the instances) and the $\boldsymbol{h}(\boldsymbol{x}_i)$ denotes the combination parameters to reconstruct the instance $\boldsymbol{x}_i$. Eq. (1) corresponds to the top panel of Figure 1, where parts are linearly combined to reconstruct the instance. Note that to fulfil the power of deep learning, the data matrix could consist of deep representations extracted by a deep neural network trained on the noisy training data.

**Approximating instance-dependent transition matrices**   Since there are computational theories [7, 59] and learning algorithms [3, 20] showing that object recognition rely on parts-based representations,

it is therefore natural to model label noise on the part level. Thus, we propose a part-dependent noise (PDN) model, where label noise depends on parts rather than the whole instances. Specifically, for each part, e.g., $W_{\cdot j}$, we assume there is a part-dependent transition matrix, e.g., $P^j \in [0,1]^{c \times c}$. Since we have $r$ parts, there are $r$ different part-dependent transition matrices, i.e., $P^j, j = 1, \ldots, r$. Similar to the idea that parts can be used to reconstruct instances, we exploit the idea that instance-dependent transition matrix can be approximated by a combination of part-dependent transition matrices, which is illustrated in the bottom panel of Figure 1.

To approximate the instance-dependent transition matrices, we need to learn the part-dependent transition matrices and the combination parameters. However, they are not identifiable because it is ill-posed to factorize the instance-dependent transition matrix into the product of part-dependent transition matrices and combination parameters. Fortunately, we could identify the part-dependent transition matrices by assuming that *the parameters for reconstructing the instance-dependent transition matrix are identical to those for reconstructing an instance*. The rational behind this assumption is that the learned parts are semantic [29], and their contributions to perceiving the instance should be similar in the contributions to understanding and annotating them [7, 3]. Let $\boldsymbol{h}(\boldsymbol{x}) \in \mathbb{R}^r$ be the combination parameters to reconstruct the instance $\boldsymbol{x}$. The instance-dependent transition matrix $T(\boldsymbol{x})$ can be approximated by

$$T(\boldsymbol{x}) \approx \sum_{j=1}^{r} \boldsymbol{h}_j(\boldsymbol{x}) P^j. \tag{2}$$

Note that $\boldsymbol{h}(\boldsymbol{x})$ can be learned via Eq. (1). The normalization constraint on the combination parameters, i.e., $\|\boldsymbol{h}(\boldsymbol{x})\|_1 = 1$, ensures that the combined matrix in the right-hand side of Eq. (2) is also a valid transition matrix, which is non-negative and the sum of each row equals one.

**Learning the part-dependent transition matrices** Note that part-dependent transition matrices in Eq. (2) are unknown. We will show that they can be learned by exploiting anchor points. The concept of anchor points was proposed in [35]. They are defined in the clean data domain, i.e., an instance $\boldsymbol{x}^i$ is an anchor point of the $i$-th clean class if $\Pr(Y = i | X = \boldsymbol{x}^i)$ is equal to one.

Let $\boldsymbol{x}^i$ be an anchor point of the $i$-th class. We have

$$\Pr(\bar{Y} = j | X = \boldsymbol{x}^i) = \sum_{k=1}^{c} \Pr(\bar{Y} = j | Y = k, X = \boldsymbol{x}^i)\Pr(Y = k | X = \boldsymbol{x}^i) = T_{ij}(\boldsymbol{x}^i), \tag{3}$$

where the first equation holds because of Law of total probability; the second equation holds because $\Pr(Y = k | X = \boldsymbol{x}^i) = 0$ for all $k \neq i$ and $\Pr(Y = i | X = \boldsymbol{x}^i) = 1$. As $[\Pr(\bar{Y} = 1 | X = \boldsymbol{x}^i), \ldots, \Pr(\bar{Y} = c | X = \boldsymbol{x}^i)]^\top$ can be unbiasedly learned [5] by exploiting the noisy training sample and the anchor point $\boldsymbol{x}^i$, Eq. (3) shows that the $i$-th row of the instance-dependent transition matrix $T(\boldsymbol{x}^i)$ can be unbiasedly learned. This sheds light on the learnability of the part-dependent transition matrices. Specifically, as shown in Figure 1, we are going to reconstruct the instance-dependent transition matrix by using a weighted combination of the part-dependent transition matrices. If the instance-dependent transition matrix[1] and combination parameters are given, learning the part-dependent transition matrices is a convex problem.

Given an anchor point $\boldsymbol{x}^i$, we can learn the $i$-th rows of the part-dependent transition matrices by matching the $i$-th row of the reconstructed transition matrix, i.e., $\sum_{j=1}^{r} \boldsymbol{h}_j(\boldsymbol{x}^i) P_{i\cdot}^j$, with the $i$-th row of the instance-dependent transition matrix, i.e., $T_{i\cdot}(\boldsymbol{x}^i)$. Since we have $r$ part-dependent transition matrices, to identify all the entries of the $i$-th rows of the part-dependent transition matrices, we need at least $r$ anchor points of the $i$-th class to build $r$ equations. Let $(\boldsymbol{x}_1^i, \ldots, \boldsymbol{x}_k^i)$ be $k$ anchor points of the $i$-th class, where $k \geq r$. We robustly learn the $i$-th rows of the part-dependent transition matrices by minimizing the reconstruction error $\sum_{l=1}^{k} \|T_{i\cdot}(\boldsymbol{x}_l^i) - \sum_{j=1}^{r} \boldsymbol{h}_j(\boldsymbol{x}_l^i) P_{i\cdot}^j\|_2^2$ instead of solving $r$ equations. Therefore, we propose the following optimization problem to learn the part-dependent transition matrices:

$$\min_{P^1, \ldots, P^r \in [0,1]^{c \times c}} \quad \sum_{i=1}^{c} \sum_{l=1}^{k} \|T_{i\cdot}(\boldsymbol{x}_l^i) - \sum_{j=1}^{r} \boldsymbol{h}_j(\boldsymbol{x}_l^i) P_{i\cdot}^j\|_2^2, \tag{4}$$

$$\text{s.t. } \|P_{i\cdot}^j\|_1 = 1, i \in \{1, \ldots, c\}, j \in \{1, \ldots, r\},$$

**Algorithm 1** Part-dependent Matrices Learning Algorithm.
---
**Input**: Noisy training sample $\mathcal{D}_\mathrm{t}$, noisy validation data $\mathcal{D}_\mathrm{v}$.
1: Train a deep model by employing the noisy data $\mathcal{D}_\mathrm{t}$ and $\mathcal{D}_\mathrm{v}$;
2: Get the deep representations of the instances by employing the trained deep network;
3. Minimize Eq. (1) to learn the parts and parameters;
4: Learn the rows of instance-dependent transition matrices by anchor points according to Eq. (3);
5: Minimize Eq. (4) to learn the part-dependent transition matrices;
6: Obtain the instance-dependent transition matrix for each instance according Eq. (2);
**Output**: $T(\boldsymbol{x})$.
---

where the sum over the index $i$ calculates the reconstruction error over all rows of transition matrices. Note that in Eq. (4), we require that anchors for each class are given. If anchor points are not available, they can be learned from the noisy data as did in [50, 35, 67].

**Implementation**  The overall procedure to learn the part-dependent transition matrices is summarized in Algorithm 1. Given only a noisy training sample set $\mathcal{D}_\mathrm{t}$, we first learn deep representations of the instances. Note that we use a noisy validation set $\mathcal{D}_\mathrm{v}$ to select the deep model. Then, we minimize Eq. (1) to learn the combination parameters. The part-dependent transition matrices are learned by minimizing Eq. (4). Finally, we use the weighted combination to get an instance-dependent transition matrix for each instance according to Eq. (2). Note that as we learn the anchor points from the noisy training data, as did in [50, 35, 67], instances that are similar to anchor points will be learned if there are no anchor points available in the training data. Then, the instance-independent transition matrix will be poorly estimated. To address this issue, we employ the slack variable $\Delta T$ in [67] to modify the instance-independent transition matrix.

## 4 Experiments

In this section, we first introduce the datasets, baselines, and implementation details used in the experiments (Section 4.1). We next conduct an ablation study to show that the proposed method is not sensitive to the number of parts (Section 4.2). Finally, we present and analyze the experimental results on synthetic and real-world noisy datasets to show the effectiveness of the proposed method (Section 4.3).

### 4.1  Experiment setup

**Datasets**  We verify the efficacy of our approach on the manually corrupted version of four datasets, i.e., *F-MNIST* [68], *SVHN* [45], *CIFAR-10* [26], *NEWS* [27], and one real-world noisy dataset, i.e., *Clothing1M* [69]. *F-MNIST* contains 60,000 training images and 10,000 test images with 10 classes. *SVHN* and *CIFAR-10* both have 10 classes of images, but the former contains 73,257 training images and 26,032 test images, and the latter contains 50,000 training images and 10,000 test images. *NEWS* contains 13,997 training texts and 6,000 test texts with 20 classes. We borrow the pre-trained word embeddings from GloVe [51] for *NEWS*. The four datasets contain clean data. We corrupted the training sets manually according to Algorithm 2. More details about this instance-dependent label noise generation approach can be found in Appendix B. IDN-$\tau$ means that the noise rate is controlled to be $\tau$. All experiments on those datasets with synthetic instance-dependent label noise are repeated five times. *Clothing1M* has 1M images with real-world noisy labels and 10k images with clean labels for testing. For all the datasets, we leave out 10% of the noisy training examples as a noisy validation set, which is for model selection. We also conduct synthetic experiments on *MNIST* [28]. Due to the space limit, we put its corresponding experimental results in Appendix C. Significance tests are conducted to show whether experimental results are statistically significant. The details for significance tests can be found in Appendix D.

**Baselines and measurements**  We compare the proposed method with the following state-of-the-art approaches: (i). CE, which trains the standard deep network with the cross entropy loss on noisy datasets. (ii). Decoupling [41], which trains two networks on samples whose the predictions from the two networks are different. (iii). MentorNet [22], Co-teaching [18], and Co-teaching+ [75]. These approaches mainly handle noisy labels by training on instances with small loss values. (iv). Joint [56], which jointly optimizes the sample labels and the network parameters. (v). DMI [70],

**Algorithm 2** Instance-dependent Label Noise Generation

---

**Input**: Clean samples $\{(\boldsymbol{x}_i, y_i)\}_{i=1}^n$; Noise rate $\tau$.
1: Sample instance flip rates $q \in \mathbb{R}^n$ from the truncated normal distribution $\mathcal{N}(\tau, 0.1^2, [0, 1])$;
2: Independently sample $w_1, w_2, \ldots, w_c$ from the standard normal distribution $\mathcal{N}(0, 1^2)$;
3: For $i = 1, 2, \ldots, n$ do
4:    $p = \boldsymbol{x}_i \times w_{y_i}$;                                        //generate instance-dependent flip rates
5:    $p_{y_i} = -\infty$;            //control the diagonal entry of the instance-dependent transition matrix
6:    $p = q_i \times softmax(p)$;      //make the sum of the off-diagonal entries of the $y_i$-th row to be $q_i$
7:    $p_{y_i} = 1 - q_i$;                                  //set the diagonal entry to be 1-$q_i$
8:    Randomly choose a label from the label space according to the possibilities $p$ as noisy label $\bar{y}_i$;
9: End for.
**Output**: Noisy samples $\{(\boldsymbol{x}_i, \bar{y}_i)\}_{i=1}^n$

---

which proposes a novel information-theoretic loss function for training deep neural networks robust to label noise. (vi). Forward [50], Reweight [35], and T-Revision [67]. These approaches utilize a class-dependent transition matrix $T$ to correct the loss function. We use the classification accuracy to evaluate the performance of each model on the clean test set. Higher classification accuracy means that the algorithm is more robust to the label noise.

**Network structure and optimization** For fair comparison, all experiments are conducted on NVIDIA Tesla V100, and all methods are implemented by PyTorch. We use a ResNet-18 network for *F-MNIST*, a ResNet-34 network for *SVHN* and *CIFAR-10*. We use a network with three convolutional layers and one fully connected layer for *NEWS*. The transition matrix $T(\boldsymbol{x})$ for each instance $\boldsymbol{x}$ can be learned according to Algorithm 1. Exploiting the transition matrices, we can bridge the class posterior probabilities for noisy and clean data. We first use SGD with momentum 0.9, weight decay $10^{-4}$, batch size 128, and an initial learning rate of $10^{-2}$ to initialize the network. The learning rate is divided by 10 at the 40th epochs and 80th epochs. We set 100 epochs in total. Then, the optimizer and learning rate are changed to Adam and $5 \times 10^{-7}$ to learn the classifier and slack variable. Note that the slack variable $\Delta T$ is initialized to be with all zero entries in the experiments. During the training, $T(\boldsymbol{x}) + \Delta T$ can be ensured to be a valid transition matrix by first projecting their negative entries to be zero and then performing row normalization. Note that we do not use any data augmentation technique in the experiments. For *Clothing1M*, we use a ResNet-50 pre-trained on ImageNet. Different from existing methods, we do not use the 50k clean training data or the 14k clean validation data but only exploit the 1M noisy data to learn the transition matrices and classifiers. Note that for real-world scenarios, it is more practical that no extra special clean data is provided to help adjust the model. After the transition matrix $T(\boldsymbol{x})$ is obtained according to the Algorithm 1, we use SGD with momentum 0.9, weight decay $10^{-3}$, batch size 32, and run with learning rate $10^{-3}$ for 10 epochs. For learning the classifier and the slack variable, Adam is used and learning rate is changed to $5 \times 10^{-7}$. Our implementation is available at https://github.com/xiaoboxia/Part-dependent-label-noise.

**Explanation** We abbreviate our proposed method of learning with the *part-dependent* transition matrices as PTD. Methods with "-F" and "-R" mean that the instance-dependent transition matrices are exploited by using the Forward [50] method and the Reweight [35] method, respectively; Methods with "-V" means that the transition matrices are revised. Details for these methods can be found in Appendix A.

### 4.2 Ablation study

We have described that how to learn part-dependent transition matrices for approximating the instance-dependent transition matrix in Section 3. To further prove that our proposed method is not sensitive to the number of parts, we perform ablation study in this subsection. The experiments are conducted on *CIFAR-10* with 50% noise rate.

In Figure 2(a), we show how well the instance-dependent transition matrix can be approximated by employing the class-dependent transition matrix and the part-dependent transition matrix. We use $\ell_1$ norm to measure the difference. For each instance, we analyze the approximation error of a specific row rather than the whole transition matrix. The reason is that we only used one row of the instance-dependent transition matrix to generate the noisy label. Specifically, given an instance with clean class label $i$ (note that we have access to clean labels for the test data to conduct evaluation), we

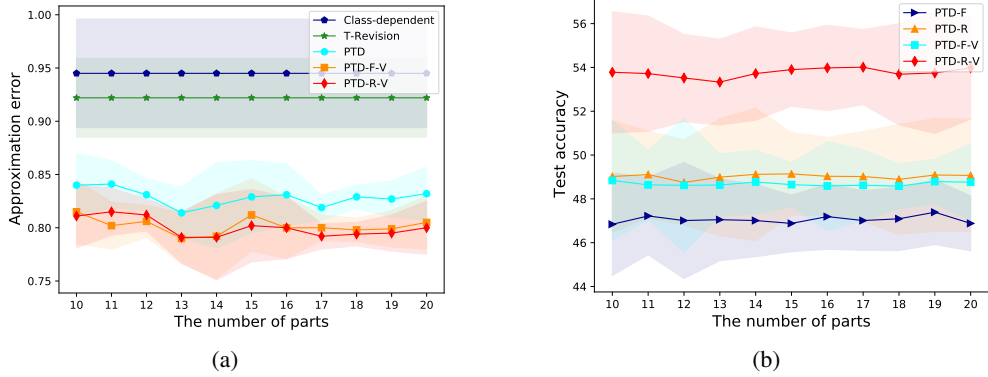

(a)                                                                                                  (b)

Figure 2: Illustration of the transition matrix approximation error and the hyperparameter sensitivity. Figure (a) illustrates how the approximation error for the instance-dependent transition matrix varies by increasing the number of parts. Figure (b) illustrates how the number of parts affects the test classification performance. The error bar for standard deviation in each figure has been shaded.

Table 1: Means and standard deviations (percentage) of classification accuracy on *F-MNIST* with different label noise levels.

|  | IDN-10% | IDN-20% | IDN-30% | IDN-40% | IDN-50% |
|---|---|---|---|---|---|
| CE | 88.54±0.31 | 88.38±0.42 | 84.22±0.35 | 68.86±0.78 | 51.42±0.66 |
| Decoupling | 89.27±0.31 | 86.50±0.35 | 85.33±0.47 | 78.54±0.53 | 57.32±2.11 |
| MentorNet | 90.00±0.34 | 87.02±0.41 | 86.02±0.82 | 80.12±0.76 | 58.62±1.36 |
| Co-teaching | 90.82±0.33 | 87.89±0.41 | 86.88±0.32 | 82.78±0.95 | 63.22±1.58 |
| Co-teaching+ | 90.92±0.51 | 89.77±0.45 | 88.52±0.45 | 83.57±1.77 | 59.32±2.77 |
| Joint | 70.24±0.99 | 56.83±0.45 | 51.27±0.67 | 44.24±0.78 | 30.45±0.45 |
| DMI | 91.98±0.62 | 90.33±0.21 | 84.81±0.44 | 69.01±1.87 | 51.64±1.78 |
| Forward | 89.05±0.43 | 88.61±0.43 | 84.27±0.46 | 70.25±1.28 | 57.33±3.75 |
| Reweight | 90.33±0.27 | 89.70±0.35 | 87.04±0.35 | 80.29±0.89 | 65.27±1.33 |
| T-Revision | 91.56±0.31 | 90.68±0.66 | 89.46±0.45 | 84.01±1.24 | 68.99±1.04 |
| PTD-F | 90.48±0.17 | 90.01±0.31 | 87.42±0.65 | 83.89±0.49 | 68.25±2.61 |
| PTD-R | 91.01±0.22 | 90.03±0.32 | 87.68±0.42 | 84.03±0.52 | 72.43±1.76 |
| PTD-F-V | 91.61±0.19 | 90.79±0.29 | 89.33±0.33 | 85.32±0.36 | 71.89±2.54 |
| PTD-R-V | **92.01±0.35** | **91.08±0.46** | **89.66±0.43** | **85.69±0.77** | **75.96±1.38** |

only exploit the $i$-th row of the instance-dependent transition matrix to flip the label from the class $i$ to another class. Note that "Class-dependent" represents the standard class-dependent transition matrix learning methods [35, 50] and "T-Revision" represents the revision methods to learn class-dependent transition matrix [67]. The Class-dependent and T-Revision methods are independent of parts. Their curves are therefore straight. We can see that the part-dependent (PTD) transition matrix can achieve much smaller approximation error than the class-dependent (part-independent) transition matrix and the results are insensitive to the number of parts. Figure 2(b) shows that the classification performance of our proposed method is robust and not sensitive to the change of the number of parts. More detailed experimental results can be found in Appendix E.

## 4.3 Comparison with the State-of-the-Arts

**Results on synthetic noisy datasets** Tables 1, 2, 3, and 4 report the classification accuracy on the datasets of *F-MNIST*, *SVHN*, *CIFAR-10*, and *NEWS*, respectively.

For *F-MNIST* and *SVHN*, in the easy cases, e.g., IDN-10% and IDN-20%, almost all methods work well. In the IDN-30% case, the advantages of PTD begin to show. We surpassed all methods obviously except for T-Revision, e.g., the classification accuracy of PTD-R-V is 1.14% higher than Co-teaching+ on *F-MNIST*, 1.33% higher than DMI on *SVHN*. When the noise rate raises, T-Revision

Table 2: Means and standard deviations (percentage) of classification accuracy on *SVHN* with different instance-dependent label noise levels.

|  | IDN-10% | IDN-20% | IDN-30% | IDN-40% | IDN-50% |
|---|---|---|---|---|---|
| CE | 90.77±0.45 | 90.23±0.62 | 86.33±1.34 | 65.66±1.65 | 48.01±4.59 |
| Decoupling | 90.49±0.15 | 90.47±0.66 | 85.27±0.34 | 82.57±1.45 | 42.56±2.79 |
| MentorNet | 90.28±0.12 | 90.37±0.37 | 86.49±0.49 | 83.75±0.75 | 40.27±3.14 |
| Co-teaching | 91.33±0.31 | 90.56±0.67 | 88.93±0.78 | 85.47±0.64 | 45.90±2.31 |
| Co-teaching+ | 93.05±1.20 | 91.05±0.82 | 85.33±2.71 | 57.24±3.77 | 42.56±3.65 |
| Joint | 86.01±0.34 | 78.58±0.72 | 76.34±0.56 | 65.14±1.72 | 46.78±3.77 |
| DMI | 93.51±1.09 | 93.22±0.62 | 91.78±1.54 | 69.34±2.45 | 48.93±2.34 |
| Forward | 90.89±0.63 | 90.65±0.27 | 87.32±0.59 | 78.46±2.58 | 46.27±3.90 |
| Reweight | 92.49±0.44 | 91.09±0.34 | 90.25±0.77 | 84.48±0.86 | 45.46±3.56 |
| T-Revision | 94.24±0.53 | 94.00±0.88 | 93.01±0.83 | 88.63±1.37 | 49.02±4.33 |
| PTD-F | 93.62±0.61 | 92.77±0.45 | 90.11±0.94 | 87.25±0.77 | 54.82±4.65 |
| PTD-R | 93.21±0.45 | 92.36±0.68 | 90.57±0.42 | 86.78±0.63 | 55.88±3.73 |
| PTD-F-V | **94.70±0.37** | **94.39±0.62** | 92.07±0.59 | 90.56±1.21 | 57.92±4.32 |
| PTD-R-V | 94.44±0.37 | 94.23±0.46 | **93.11±0.78** | **90.64±0.98** | **58.09±2.57** |

Table 3: Means and standard deviations (percentage) of classification accuracy on *CIFAR-10* with different label noise levels.

|  | IDN-10% | IDN-20% | IDN-30% | IDN-40% | IDN-50% |
|---|---|---|---|---|---|
| CE | 74.49±0.29 | 68.21±0.72 | 60.48±0.62 | 49.84±1.27 | 38.86±2.71 |
| Decoupling | 74.09±0.78 | 70.01±0.66 | 63.05±0.65 | 44.27±1.91 | 38.63±2.32 |
| MentorNet | 74.45±0.66 | 70.56±0.34 | 65.42±0.79 | 46.22±0.98 | 39.89±2.62 |
| Co-teaching | 76.99±0.17 | 72.99±0.45 | 67.22±0.64 | 49.25±1.77 | 42.77±3.41 |
| Co-teaching+ | 74.27±1.20 | 71.07±0.77 | 64.77±0.58 | 47.73±2.32 | 39.47±2.14 |
| Joint | 76.89±0.37 | 73.89±0.34 | 69.03±0.79 | 54.75±5.98 | 44.72±7.72 |
| DMI | 75.02±0.45 | 69.89±0.33 | 61.88±0.64 | 51.23±1.18 | 41.45±1.97 |
| Forward | 73.45±0.23 | 68.99±0.62 | 60.21±0.75 | 47.17±2.96 | 40.75±2.09 |
| Reweight | 74.55±0.23 | 68.42±0.75 | 62.58±0.46 | 50.12±0.96 | 41.08±2.45 |
| T-Revision | 74.61±0.39 | 69.32±0.64 | 64.09±0.37 | 50.38±0.87 | 42.57±3.27 |
| PTD-F | 76.01±0.45 | 73.45±0.62 | 65.25±0.84 | 49.88±0.85 | 46.88±1.25 |
| PTD-R | 78.71±0.22 | 75.02±0.73 | 71.86±0.42 | 56.15±0.45 | 49.07±2.56 |
| PTD-F-V | 76.29±0.38 | 73.88±0.61 | 69.01±0.47 | 50.43±0.62 | 48.76±2.01 |
| PTD-R-V | **79.01±0.20** | **76.05±0.53** | **72.28±0.49** | **58.62±0.88** | **53.98±2.34** |

is gradually defeated. In the IDN-30% case, the classification accuracy of PTD-R-V is 1.68% and 2.01% higher than T-Revision on *SVHN* and *CIFAR-10* respectively. Finally, in the hardest case, i.e., IDN-50%, the superiority of PTD widens the gap of performance. The classification accuracy of PTD-R-V is 6.97% and 9.07% higher than the best baseline method.

For *CIFAR-10*, the algorithms with the assist of PTD overtake the other methods with clear gaps. From IDN-10% to IDN-50% case, the advantages of our proposed method increase with the increasing of the noise rate. In the 10% and 20% cases, the performance of PTD-R-V is outstanding, i.e., the classification accuracy is 2.02% and 2.16% higher than the best baseline Joint. In the 30% and 40% case, the gap is expanded to 3.25% and 3.87%. Lastly, in the 50% case, PTD-R-V outperforms state-of-the-art methods by almost 10% of classification accuracy.

For *NEWS*, the proposed method PTD-R-V consistently outperforms all the baseline methods. From IDN-10% to IDN-40% case, PTD-R-V clearly surpasses the best baseline T-Revision. In the hardest case, i.e., IDN-50%, PTD-R-V outperforms all T-Revision by more than 3% of test accuracy.

To sum up, the synthetic experiments reveal that our method is powerful in handling *instance-dependent* label noise particularly in the situation of high noise rates.

**Results on real-world datasets**    The proposed method outperforms the baselines as shown in Table 5, where the highest accuracy is bold faced. The comparison denotes that, the noise model of

Table 4: Means and standard deviations (percentage) of classification accuracy on *NEWS* with different label noise levels.

|  | IDN-10% | IDN-20% | IDN-30% | IDN-40% | IDN-50% |
|---|---|---|---|---|---|
| CE | 69.58±0.42 | 66.80±0.36 | 63.11±0.74 | 58.37±0.88 | 54.75±1.62 |
| Decoupling | 69.35±0.41 | 65.32±0.43 | 58.75±0.84 | 51.63±0.77 | 43.05±1.52 |
| MentorNet | 69.03±0.35 | 66.92±0.54 | 62.87±1.31 | 54.35±1.21 | 48.35±1.45 |
| Co-teaching | 69.37±0.29 | 67.99±0.76 | 64.15±0.89 | 56.36±0.71 | 52.32±1.03 |
| Co-teaching+ | 69.35±0.73 | 64.03±0.91 | 56.37±0.61 | 41.88±1.74 | 10.78±5.87 |
| Joint | 69.73±0.51 | 67.45±0.49 | 64.54±0.74 | 60.67±0.83 | 56.72±2.10 |
| DMI | 70.35±0.62 | 68.01±0.45 | 64.28±0.61 | 60.73±0.62 | 56.33±1.35 |
| Forward | 69.24±0.45 | 66.01±0.55 | 62.07±0.58 | 56.33±0.71 | 53.25±1.43 |
| Reweight | 70.25±0.30 | 68.42±0.77 | 65.05±0.93 | 59.37±1.32 | 57.31±3.51 |
| T-Revision | 70.72±0.32 | 69.91±0.49 | 67.28±0.81 | 61.78±0.99 | 59.29±2.07 |
| PTD-F | 70.01±0.47 | 66.78±0.68 | 62.16±0.77 | 59.54±0.63 | 53.63±1.31 |
| PTD-R | 71.03±0.45 | 70.02±0.53 | 68.32±0.72 | 62.37±0.45 | 62.01±1.21 |
| PTD-F-V | 70.27±0.28 | 66.81±0.48 | 62.80±0.76 | 59.71±0.46 | 54.23±1.17 |
| PTD-R-V | **71.92±0.34** | **71.33±0.42** | **69.01±0.85** | **63.17±0.58** | **62.77±0.98** |

Table 5: Classification accuracy on *Clothing1M*. In the experiments, only noisy samples are exploited to train and validate the deep model.

| CE | Decoupling | MentorNet | Co-teaching | Co-teaching+ | Joint | DMI |
|---|---|---|---|---|---|---|
| 68.88 | 54.53 | 56.79 | 60.15 | 65.15 | 70.88 | 70.12 |
| Forward | Reweight | T-Revision | PTD-F | PTD-R | PTD-F-V | PTD-R-V |
| 69.91 | 70.40 | 70.97 | 70.07 | 71.51 | 70.26 | **71.67** |

*Clothing1M* dataset is more likely to be *instance-dependent* noise, and our proposed method can better model instance-dependent noise than other methods.

## 5   Conclusion

In this paper, we focus on learning with *instance-dependent* label noise, which is a more general case of label noise but lacking understanding and learning. Inspired by parts-based learning, we exploit *part-dependent* transition matrix to approximate *instance-dependent* transition matrix, which is intuitive and learnable. Specifically, we first learn the parts of instances using all training examples. Then, we learn the part-dependent transition matrices by exploiting anchor points. Lastly, the instance-dependent transition matrix can be well approximated by a combination of the part-dependent transition matrices. Experimental results show our proposed method consistently outperforms existing methods, especially for the case of high-level noise rates. In future, we can extend the work in the following aspects. First, we can incorporate some prior knowledge of transition matrix and parts (e.g., sparsity), which improves parts-based learning. Second, we can introduce slack variables to modify the parameters for combination.

## Broader Impact

*Instance-dependent* label noise is ubiquitous in the era of big data, which poses huge reliability threats for the traditional supervised learning algorithms. The instance-dependent label noise is more general and more realistic than *instance-independent* label noise, but is hard to learn without any assumption. How to model such noise and reduce its side-effect should be considered by both research and industry communities. This research copes with *instance-dependent* label noise based on the *part-dependence* assumption. This assumption is milder and more practical. It is also supported by lots of evidences as stated in the paper. Outcomes of this research will promote the understanding of this kind of label noise and largely fills the gap between instance-independent and instance-dependent

transition matrices. Open source algorithms and codes will benefit science, society, and the economy internationally through the applications to analyzing social, business, and health data.

The research may greatly benefit practitioners in industry communities, where large amounts of noisily labeled data are available. However, currently, the majority of machine learning applications are designed to fit high-quality labeled data. This research will improve tolerance for the errors of annotation and make cheap datasets with label noise be used effectively. However, inevitably, this research may have a negative impact on the jobs of annotators.

The proposed method exploits the *part-dependent* transition matrices to approximate the *instance-dependent* transition matrix. If the *part-dependent* transition matrices are poorly learned, the *instance-dependent* transition matrix will be inaccurate. The classification performance of models therefore may be compromised.

The proposed method does not leverage any bias in the data.

## Acknowledgments

TLL was supported by Australian Research Council Project DE-190101473 and DP-180103424. BH was supported by the RGC Early Career Scheme No. 22200720, NSFC Young Scientists Fund No. 62006202, HKBU Tier-1 Start-up Grant, and HKBU CSD Start-up Grant. NNW was supported by National Natural Science Foundation of China under Grant 61922066 and Grant 61876142. DCT was supported by Project FL-170100117, DP-180103424, and IH-180100002. GN and MS were supported by JST AIP Acceleration Research Grant Number JPMJCR20U3, Japan. The authors would give special thanks to Pengqian Lu for helpful discussions and comments. The authors thank the reviewers and the meta-reviewer for their helpful and constructive comments on this work.

## Footnotes

[1]Note that according to (3), given an anchor point $\boldsymbol{x}_i$, the $i$-th row of its instance-dependent transition matrix can be learned and thus available.

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
