[Supplementary Material]

# Supplementary to "Part-dependent Label Noise: Towards Instance-dependent Label Noise"

## A    How to learn robust classifiers by exploiting part-dependent transition matrices

For those who are not familiar with how to use the transition matrix to learn robust classifiers, in this supplementary material, we will provide how to learn robust classifiers by exploiting part-dependent transition matrices.

We begin by introducing notation. Let $D$ be the distribution of the variables $(X, Y)$, $\bar{D}$ the distribution of the variables $(X, \bar{Y})$. Let $S = \{(\boldsymbol{x}_i, y_i)\}_{i=1}^n$ be i.i.d. samples drawn from the distribution $D$, $\bar{S} = \{(\boldsymbol{x}_i, \bar{y}_i)\}_{i=1}^n$ i.i.d. samples drawn from the distribution $\bar{D}$, and $c$ the size of label classes.

The aim of multi-class classification is to learn a classifier $f$ that can assign labels for given instances. The classifier $f$ is of the following form: $f(\boldsymbol{x}) = \arg\max_{i \in \{1,2,...,c\}} g_i(\boldsymbol{x})$, where $g_i(\boldsymbol{x})$ is an estimate of $\Pr(Y = i | X = \boldsymbol{x})$. *Expected risk* of employing $f$ is defined as

$$R(f) = \mathbb{E}_{(X,Y) \sim D}[\ell(f(X), Y)]. \tag{1}$$

The optimal classifier to learn is the one that minimizes the risk $R(f)$. Due to the distribution $D$ is usually unknown, the optimal classifier is approximated by the minimizer of the *empirical risk*:

$$R_n(f) = \frac{1}{n} \sum_{i=1}^n \ell(f(\boldsymbol{x}_i), y_i). \tag{2}$$

Given only the noisy training samples $\{(\boldsymbol{x}_i, \bar{y}_i)\}_{i=1}^n$, the noisy version of the empirical risk is defined as:

$$\bar{R}_n(f) = \frac{1}{n} \sum_{i=1}^n \ell(f(\boldsymbol{x}_i), \bar{y}_i). \tag{3}$$

In the main paper (Section 3), we show how to approximate *instance-dependent* transition matrix by exploiting *part-dependent* transition matrices. For an instance $\boldsymbol{x}$, according to the definition of *instance-dependent* transition matrix, we have that $\Pr(\bar{\mathbf{Y}} | X = \boldsymbol{x}) = T^\top(\boldsymbol{x}) \Pr(\mathbf{Y} | X = \boldsymbol{x})$, we let

$$\bar{h}(\boldsymbol{x}) = \arg\max_{i \in \{1,2,...,c\}} (T^\top(\boldsymbol{x}) g)_i(\boldsymbol{x}). \tag{4}$$

The empirical risk of our PTD-F algorithm is defined as:

$$\bar{R}_n(\bar{h}) = \frac{1}{n} \sum_{i=1}^n \ell(\bar{h}(\boldsymbol{x}_i), \bar{y}_i). \tag{5}$$

By employing the *importance reweighting* technique [1, 2, 4], the empirical risk of our PTD-R algorithm is defined as:

$$\bar{R}_n(f, \bar{h}) = \frac{1}{n} \sum_{i=1}^n \frac{g_{\bar{y}_i}(\boldsymbol{x}_i)}{\bar{h}_{\bar{y}_i}(\boldsymbol{x}_i)} \ell(f(\boldsymbol{x}_i), \bar{y}_i). \tag{6}$$

---
**Algorithm 2** Instance-dependent Label Noise Generation
---
**Input**: Clean samples $\{(\boldsymbol{x}_i, y_i)\}_{i=1}^n$; Noise rate $\tau$.
1: Sample instance flip rates $q \in \mathbb{R}^n$ from the truncated normal distribution $\mathcal{N}(\tau, 0.1^2, [0, 1])$;
2: Independently sample $w_1, w_2, \ldots, w_c$ from the standard normal distribution $\mathcal{N}(0, 1^2)$;
3: For $i = 1, 2, \ldots, n$ do
4:    $p = \boldsymbol{x}_i \times w_{y_i}$;                                               //generate instance-dependent flip rates
5:    $p_{y_i} = -\infty$;               //control the diagonal entry of the instance-dependent transition matrix
6:    $p = q_i \times softmax(p)$;       //make the sum of the off-diagonal entries of the $y_i$-th row to be $q_i$
7:    $p_{y_i} = 1 - q_i$;                                //set the diagonal entry to be 1-$q_i$
8:    Randomly choose a label from the label space according to the possibilities $p$ as noisy label $\bar{y}_i$;
9: End for.
**Output**: Noisy samples $\{(\boldsymbol{x}_i, \bar{y}_i)\}_{i=1}^n$
---

Here, $g_j(\boldsymbol{x})$ is an estimate for $\Pr(Y = j|\boldsymbol{x})$ and $h_j(\boldsymbol{x})$ is an estimate for $\Pr(\bar{Y} = j|\boldsymbol{x})$.

When the slack variable $\Delta T$ is introduced to modify the instance-dependent transition matrices, reviewing Eq. (4), we replace $T(\boldsymbol{x})$ with $T(\boldsymbol{x}) + \Delta T$ to get $\bar{h}'(\boldsymbol{x})$, i.e.,

$$\bar{h}'(\boldsymbol{x}) = \arg \max_{i \in \{1,2,\ldots,c\}} (T(\boldsymbol{x}) + \Delta T)^\top g)_i(\boldsymbol{x}). \tag{7}$$

Then the empirical risks of PTD-F-V and PTD-R-V are defined as $\bar{R}_n(\bar{h}')$ and $\bar{R}_n(f, \bar{h}')$, i.e.,

$$\bar{R}_n(\bar{h}') = \frac{1}{n} \sum_{i=1}^n \ell(\bar{h}'(\boldsymbol{x}_i), \bar{y}_i). \tag{8}$$

and

$$\bar{R}_n(f, \bar{h}') = \frac{1}{n} \sum_{i=1}^n \frac{g_{\bar{y}_i}(\boldsymbol{x}_i)}{\bar{h}'_{\bar{y}_i}(\boldsymbol{x}_i)} \ell(f(\boldsymbol{x}_i), \bar{y}_i). \tag{9}$$

To learn noise robust classifiers under noisy supervision, we minimize the empirical risk of PTD-F, PTD-R, PTD-F-V, and PTD-R-V, respectively.

## B    Instance-dependent Label Noise Generation

Note that it is more realistic that different instances have different flip rates. Without constraining different instances to have a same flip rate, it is more challenging to model the label noise and train robust classifiers. In Step 1, in order to control the global flip rate as $\tau$ but without constraining all of the instances to have a same flip rate, we sample their flip rates from a truncated normal distribution $\mathcal{N}(\tau, 0.1^2, [0, 1])$. Specifically, this distribution limits the flip rates of instances in the range $[0, 1]$. Their mean and standard deviation are equal to the mean $\tau$ and the standard deviation 0.1 of the selected truncated normal distribution respectively.

In Step 2, we sample parameters $w_1, w_2, \ldots, w_c$ from the standard normal distribution for generating *instance-dependent* label noise. The dimensionality of each parameter is $d \times c$, where $d$ denotes the dimensionality of the instance. Learning these parameters is critical to model *instance-dependent* label noise. However, it is hard to identify these parameters without any assumption.

Note that an instance with clean label $y$ will be flipped only according to the $y$-th row of the transition matrix. Thus, in Steps 4 to 7, we only use the $y_i$-th row of the *instance-dependent* transition matrix for the instance $\boldsymbol{x}_i$. Specifically, Steps 5 and 7 are to ensure the diagonal entry of the $y_i$-th row is 1-$q_i$. Step 6 is to ensure that the sum of the off-diagonal entries is $q_i$.

## C    Experiments complementary on synthetic noisy datasets

In the main paper (Section 4), we present the experimental results on four synthetic noisy datasets, i.e., *F-MNIST*, *SVHN*, *CIFAR-10*, and *NEWS*. In this supplementary material, we provide the experimental

Table 1: Means and standard deviations (percentage) of classification accuracy on *MNIST* with different label noise levels.

|            | IDN-10%         | IDN-20%         | IDN-30%         | IDN-40%         | IDN-50%         |
|------------|-----------------|-----------------|-----------------|-----------------|-----------------|
| CE          | 98.24±0.07     | 98.21±0.06     | 96.78±0.12     | 93.76±0.18     | 79.69±4.35     |
| Decoupling  | 96.63±0.12     | 96.62±0.22     | 92.73±0.36     | 90.34±0.33     | 80.56±2.67     |
| MentorNet   | 97.45±0.11     | 97.21±0.13     | 92.88±0.31     | 88.23±1.65     | 80.02±1.71     |
| Co-teaching | 97.56±0.12     | 97.32±0.15     | 94.81±0.24     | 92.45±0.59     | 83.30±1.37     |
| Co-teaching+| 98.32±0.07     | 98.07±0.12     | 96.70±0.35     | 94.37±0.48     | 82.97±1.11     |
| Joint       | 98.53±0.06     | 98.17±0.14     | 96.51±0.17     | 93.07±0.62     | 83.72±3.22     |
| DMI         | 98.63±0.04     | 98.40±0.11     | 97.75±0.21     | 96.45±0.23     | 87.52±1.03     |
| Forward     | 97.23±0.15     | 96.87±0.15     | 95.01±0.27     | 90.30±0.61     | 77.42±3.28     |
| Reweight    | 98.21±0.07     | 97.99±0.13     | 96.96±0.14     | 94.55±0.67     | 80.87±4.14     |
| T-Revision  | 98.49±0.06     | 98.39±0.09     | 97.55±0.14     | 96.50±0.31     | 84.71±3.47     |
| PTD-F       | 98.55±0.05     | 97.92±0.27     | 97.34±0.11     | 94.67±0.83     | 84.01±2.11     |
| PTD-R       | 98.22±0.10     | 98.12±0.17     | 97.06±0.13     | 94.75±0.54     | 82.72±2.04     |
| PTD-F-V     | **98.71±0.05** | **98.46±0.11** | 97.77±0.09     | 96.07±0.45     | 88.55±1.96     |
| PTD-R-V     | 98.66±0.03     | 98.43±0.15     | **97.81±0.23** | **96.73±0.20** | **88.67±1.25** |

results on another synthetic noisy dataset *MNIST*. *MNIST* contains 60,000 training images and 10,000 test images with 10 classes. We use a LeNet-5 network for it. The detailed experimental results are shown in Table 1. The classification performance shows that our proposed method is more robust than the baseline methods when coping with *instance-dependent* label noise.

## D    The details of significance tests

We exploit significance tests to show whether all experimental results are statistically significant. The *p*-values are obtained with two independent samples t-test [3]. Note that small *p*-values reflect the performance of the proposed method is significantly better than the performance of the baselines. The proposed method PTD-R-V achieves the best classification performance in almost all cases. We thus conduct significance tests to compare the baselines with PTD-R-V. The results of significance tests are presented in Table 2. We can see that almost all results are statistically significant.

## E    The experimental results of ablation study

In Section 4.2, we have shown that our proposed method is insensitive to the number of parts. Due the space limit, we only provide the illustration by exploiting the figures. In this supplementary material, more detailed results including means and standard deviations of approximation error and classification accuracy about the ablation study are shown in Table 3 and Table 4.

## F    Visualization of parts

Note that to make use of the power of deep learning, in the main paper, the data matrix used for factorization consists of deep representations extracted by a deep network. We learn parts and parts-based representations (new representations) by applying NMF to this data matrix. Although deep representations contain semantic information, it is not easy to visualize these parts obtained from the deep representations directly. We propose to approximate and visualize the parts of the deep representations by studying their corresponding parts of the original observations. Intuitively, let the NMF of the deep representations and the original observations to have the same parts-based representations, the obtained parts from the two factorizations should be corresponding to each other. The obtained parts for *MNIST* and *F-MNIST* are presented in Figure 1 and Figure 2. Note that the datasets, i.e., *SVHN*, *CIFAR-10*, and *Clothing1M*, are also used to verify the effectiveness of the proposed method. However, the instances in these datasets contain three channels (i.e., RGB channels). It is hard to properly visualize the parts of the deep representations by finding their corresponding parts of the original observations.

Table 2: The results of significant tests ($p$-value) on five synthetic noisy datasets with different noise levels.

| Dataset | Method | IDN-10% | IDN-20% | IDN-30% | IDN-40% | IDN-50% |
|---|---|---|---|---|---|---|
| MNIST | CE | 0.0000 | 0.0187 | 0.0000 | 0.0000 | 0.0152 |
| | Decoupling | 0.0000 | 0.0000 | 0.0000 | 0.0000 | 0.0002 |
| | MentorNet | 0.0000 | 0.0000 | 0.0000 | 0.0000 | 0.0004 |
| | Co-teaching | 0.0000 | 0.0000 | 0.0000 | 0.0000 | 0.0001 |
| | Co-teaching+ | 0.0000 | 0.0005 | 0.0009 | 0.0000 | 0.0000 |
| | Joint | 0.0036 | 0.0202 | 0.0000 | 0.0000 | 0.0112 |
| | DMI | 0.0016 | 0.4489 | 0.0157 | 0.1732 | 0.4395 |
| | Forward | 0.0000 | 0.0000 | 0.0000 | 0.0000 | 0.0000 |
| | Reweight | 0.0000 | 0.0044 | 0.0001 | 0.0011 | 0.0025 |
| | T-Revision | 0.0050 | 0.8045 | 0.0058 | 0.2130 | 0.0695 |
| F-MNIST | CE | 0.0000 | 0.0001 | 0.0000 | 0.0000 | 0.0000 |
| | Decoupling | 0.0000 | 0.0000 | 0.0000 | 0.0000 | 0.0000 |
| | MentorNet | 0.0000 | 0.0000 | 0.0003 | 0.0001 | 0.0000 |
| | Co-teaching | 0.0000 | 0.0000 | 0.0000 | 0.0173 | 0.0000 |
| | Co-teaching+ | 0.0737 | 0.0023 | 0.0030 | 0.4358 | 0.0000 |
| | Joint | 0.0000 | 0.0000 | 0.0000 | 0.0000 | 0.0000 |
| | DMI | 0.4281 | 0.0068 | 0.0000 | 0.0000 | 0.0000 |
| | Forward | 0.0001 | 0.0002 | 0.0000 | 0.0000 | 0.0000 |
| | Reweight | 0.0000 | 0.0041 | 0.0000 | 0.0001 | 0.0001 |
| | T-Revision | 0.9522 | 0.1335 | 0.1626 | 0.0931 | 0.0002 |
| SVHN | CE | 0.0000 | 0.0000 | 0.0000 | 0.0000 | 0.0012 |
| | Decoupling | 0.0000 | 0.0000 | 0.0000 | 0.0005 | 0.0000 |
| | MentorNet | 0.0000 | 0.0000 | 0.0001 | 0.0000 | 0.0000 |
| | Co-teaching | 0.0000 | 0.0000 | 0.0001 | 0.0000 | 0.0000 |
| | Co-teaching+ | 0.0001 | 0.0000 | 0.0000 | 0.0000 | 0.0005 |
| | Joint | 0.0000 | 0.0000 | 0.0000 | 0.0000 | 0.0001 |
| | DMI | 0.0068 | 0.0385 | 0.6901 | 0.0000 | 0.0002 |
| | Forward | 0.0000 | 0.0000 | 0.0002 | 0.0000 | 0.0000 |
| | Reweight | 0.0001 | 0.0139 | 0.0031 | 0.0018 | 0.0002 |
| | T-Revision | 0.2258 | 0.3116 | 0.5436 | 0.0471 | 0.0228 |
| CIFAR-10 | CE | 0.0000 | 0.0000 | 0.0000 | 0.0000 | 0.0000 |
| | Decoupling | 0.0001 | 0.0000 | 0.0000 | 0.0000 | 0.0000 |
| | MentorNet | 0.0000 | 0.0000 | 0.0000 | 0.0000 | 0.0000 |
| | Co-teaching | 0.0000 | 0.0000 | 0.0000 | 0.0000 | 0.0000 |
| | Co-teaching+ | 0.0002 | 0.0001 | 0.0000 | 0.0001 | 0.0000 |
| | Joint | 0.0000 | 0.0000 | 0.0083 | 0.0704 | 0.0638 |
| | DMI | 0.0000 | 0.0000 | 0.0000 | 0.0000 | 0.0000 |
| | Forward | 0.0000 | 0.0000 | 0.0000 | 0.0000 | 0.0000 |
| | Reweight | 0.0000 | 0.0000 | 0.0000 | 0.0000 | 0.0000 |
| | T-Revision | 0.0000 | 0.0000 | 0.0000 | 0.0000 | 0.0013 |
| NEWS | CE | 0.0000 | 0.0000 | 0.0000 | 0.0000 | 0.0002 |
| | Decoupling | 0.0000 | 0.0000 | 0.0000 | 0.0000 | 0.0000 |
| | MentorNet | 0.0000 | 0.0001 | 0.0000 | 0.0000 | 0.0000 |
| | Co-teaching | 0.0000 | 0.0027 | 0.0125 | 0.0000 | 0.0000 |
| | Co-teaching+ | 0.0000 | 0.0001 | 0.0000 | 0.0000 | 0.0000 |
| | Joint | 0.0000 | 0.0000 | 0.0001 | 0.0025 | 0.0008 |
| | DMI | 0.0004 | 0.0000 | 0.0004 | 0.0032 | 0.0001 |
| | Forward | 0.0000 | 0.0000 | 0.0000 | 0.0000 | 0.0000 |
| | Reweight | 0.0000 | 0.0001 | 0.0008 | 0.0021 | 0.0108 |
| | T-Revision | 0.0010 | 0.0006 | 0.0040 | 0.0052 | 0.0285 |

Table 3: Means and standard deviations of approximation error on *CIFAR-10* with 50% label noise level.

|  | Class-dependent | T-Revision | PTD | PTD-F-V | PTD-R-V |
|---|---|---|---|---|---|
| $r$=10 | 0.945±0.051 | 0.922±0.037 | 0.840±0.030 | 0.815±0.011 | 0.811±0.020 |
| $r$=11 | 0.945±0.051 | 0.922±0.037 | 0.841±0.022 | 0.802±0.010 | 0.815±0.011 |
| $r$=12 | 0.945±0.051 | 0.922±0.037 | 0.831±0.015 | 0.806±0.014 | 0.812±0.014 |
| $r$=13 | 0.945±0.051 | 0.922±0.037 | 0.814±0.024 | 0.790±0.019 | 0.791±0.017 |
| $r$=14 | 0.945±0.051 | 0.922±0.037 | 0.821±0.040 | 0.792±0.022 | 0.791±0.016 |
| $r$=15 | 0.945±0.051 | 0.922±0.037 | 0.829±0.034 | 0.812±0.017 | 0.802±0.025 |
| $r$=16 | 0.945±0.051 | 0.922±0.037 | 0.831±0.029 | 0.800±0.018 | 0.800±0.020 |
| $r$=17 | 0.945±0.051 | 0.922±0.037 | 0.819±0.012 | 0.800±0.011 | 0.792±0.013 |
| $r$=18 | 0.945±0.051 | 0.922±0.037 | 0.829±0.011 | 0.798±0.012 | 0.794±0.017 |
| $r$=19 | 0.945±0.051 | 0.922±0.037 | 0.827±0.017 | 0.799±0.013 | 0.795±0.018 |
| $r$=20 | 0.945±0.051 | 0.922±0.037 | 0.832±0.025 | 0.805±0.021 | 0.800±0.015 |

Table 4: Means and standard deviations (percentage) of classifation accuracy on *CIFAR-10* with 50% label noise level.

|  | PTD-F | PTD-R | PTD-F-V | PTD-R-V |
|---|---|---|---|---|
| $r$=10 | 46.84±2.34 | 49.02±2.55 | 48.84±2.74 | 53.78±2.77 |
| $r$=11 | 47.22±1.77 | 49.11±1.98 | 48.64±1.58 | 53.72±2.63 |
| $r$=12 | 47.01±2.65 | 48.75±1.95 | 48.62±3.05 | 53.52±1.99 |
| $r$=13 | 47.05±1.87 | 48.99±2.67 | 48.63±1.42 | 53.33±1.96 |
| $r$=14 | 47.01±1.65 | 49.12±3.02 | 48.77±1.46 | 53.72±2.13 |
| $r$=15 | 46.88±1.29 | 49.14±1.89 | 48.65±1.01 | 53.90±1.67 |
| $r$=16 | 47.19±1.49 | 49.03±1.78 | 48.59±2.03 | 53.98±1.95 |
| $r$=17 | 47.01±1.36 | 49.02±2.06 | 48.62±1.62 | 54.01±1.72 |
| $r$=18 | 47.09±1.45 | 48.89±2.51 | 48.58±1.03 | 53.69±2.31 |
| $r$=19 | 47.39±1.48 | 49.09±2.58 | 48.79±1.01 | 53.75±2.77 |
| $r$=20 | 46.88±1.25 | 49.07±2.56 | 48.76±1.75 | 53.98±2.34 |

Figure 1: Visualization of parts for *MNIST*.

Figure 2: Visualization of parts for *F-MNIST*.