[Reviews · NeurIPS 2020]

Review 1

Summary and Contributions: In this paper, a new method for approximating the instance-dependent label noise by using parts-dependent label noise is proposed. The method is built upon the assumption that an instance-dependent transition matrix for an instance is constructed by a combination of the transition matrices for the parts of the instance. The motivation for this is based on the conjecture that instances can be approximately reconstructed by a combination of parts. Here, The transition matrices for the parts are learned based on anchor points. Finally the method is evaluated on some small dataset, Fashion-MNIST, SVHN, and CIFAR-10 to show the superiority of the methods in comparison to SOTA.

Strengths: + a new and interesting method for approximating the instance-dependent label noise. + rigorous analysis and evaluation on small datasets to demonstrate the superiority of the proposed method in comparison to SOTA.

Weaknesses: 1- The main assumption in this method for part-dependent label noise is not realistic to me. The method learns the parts using the anchor points, and these parts are used for all the samples belonging to the same class for representing them. However, there are images in the same classes (e.g., hard samples) that may not be represented properly using the combination of the learned parts from the anchor points. Considering the fact that anchor points are usually easy samples as the model has a high confidence regarding their labels and may not contain all the information of other samples belonging to the same class. However, I think the method only has been evaluated on the small datasets which may not show this. 2- In the paper it is mentioned that “the data matrix could consist of deep representations extracted by a deep neural network trained on the noisy training data.” When the deep model is trained using the noisy labels, I was thinking that the features or data matrix is not the accurate or reliable features that we could define a loss function like EQ. (1). And therefore “W, matrix of parts, and h(xi) may be poorly estimated. 3- As a minor point: “The parameters for reconstructing the instance-dependent transition matrix are identical to those for reconstructing an instance.” This conjecture has been used for estimating h(xi) parameters in the model which is an important parameter.” Although there are some references in the paper for supporting this conjecture, could you please be more specific in references and elaborate on this. 4- Finally, it would be very nice to visualize parts for some classes of datasets used in the experiments, like the one shown in Figure. 1. This can better contextualize the power of the proposed method for reconstructing images using the parts learned by the proposed model.

Correctness: Yes, to some extent.

Clarity: Yes, the paper is well-written. The concepts are explained precisely.

Relation to Prior Work: Yes.

Reproducibility: Yes

Additional Feedback: Please refer to weaknesses. -------------------------------------------------------------- after rebuttal: I read the authors' feedback and the other reviewers' comments. I am happy that the authors release my main concerns that I mentioned in my reviews. For example, the early stopping technique on validation set for alleviating the effect of memorization in deep models. I hope that the authors visualize parts for some classes of datasets and also improve the clarity of the conjecture used in their work in the camera ready. Overall, I think this is a good piece of work for approximating the instance-dependent label noise. Therefore, I change my score from marginal accept to a clear accept.


Review 2

Summary and Contributions: This paper investigates the problem of multi-class learning with instance-dependent label noise. The proposed method of this paper is motivated by the human cognition that annotators are more likely to annotate instances based on the parts rather than the whole instances. Each instance can be approximately reconstructed by a combination of parts. So this paper first proposes to use non-negative matrix factorization to obtain the reconstruction weights. Then, this paper exploits the anchor points to learn the parts-dependent transition matrices. Finally, the instance-dependent transition matrix (of each instance) could be reconstructed by the parts-dependent transition matrices and reconstruction weights. Overall, I think this is a good paper, because the problem of multi-class learning with instance-dependent label noise was rarely studied and this paper proposes an interesting and decent method to solve this problem. I feel that the most interesting point is that although we cannot directly obtain the instance-dependent transition matrix for each instance, but we can still exploit the anchor points to directly obtain the instance-dependent transition matrix for each anchor point. In this way, the parts-dependent transition matrices can be learned by using those transition matrices of anchor points and the reconstruction weights.

Strengths: 1. The problem of multi-class learning with instance-dependent label noise was rarely studied. 2. This paper proposes an interesting and decent method to solve this problem. 3. Experiments demonstrate the effectiveness of the proposed method.

Weaknesses: 1. I feel that the presentation of the subsection "Learning the parts-dependent transition matrices" could be further improved, because it seems that the key point is not so clear. 2. For the experimental results, it would be better to show whether the performance gap is significant or not.

Correctness: Yes

Clarity: Yes

Relation to Prior Work: Yes

Reproducibility: Yes

Additional Feedback: Please see the weaknesses. ======= I have read the comments and the rebuttal. The authors said that they conducted significant tests and all results were statistically significant, and will provide experimental results in the final version. So I keep my rating (accept) unchanged.


Review 3

Summary and Contributions: Learning with instance-dependent label noise is a vital part of label noise learning. However, it’s hard to model such realistic noise only exploiting noisy data. This paper assumed that the instance-dependent label noise depends only on parts of instances, which was motivated by annotators annotate instances based on their parts. The authors further proposed to approximate the instance-dependent transition matrix with a combination of parts-dependent transition matrices. Extensive experimental results on syntheic and real-world datasets verified the effectiveness of the proposed method. Overall, this paper makes a significant contribution to learning with instance-dependent label noise.

Strengths: 1. The proposed method is very novel and well motivated, which makes a lot of sense in learning with instance-dependent label noise. 2. The explanation of the proposed approach is sufficient and clear. Besides, the organization and logic of this paper make it easy to understand this idea. 3. The description of experimental settings is detailed. The experimental results on benchmark datasets are very convincing. Besides, the authors provided a detailed analysis of experimental results.

Weaknesses: 1. Although the idea makes sense, the proposed method contains multiple stages according to algorithm flow. It will be better to achieve end-to-end training. 2. In addition to the experimental results, this paper needs some more detailed explanations intuitively or theoretically as to why the other state-of-the-art methods don’t work for learning from instance-dependent label noise.

Correctness: Yes, the claims and method are correct. The empirical methodology is correct.

Clarity: Yes, the paper is well written.

Relation to Prior Work: Yes, it is clearly discussed how this work differs from previous contributions.

Reproducibility: Yes

Additional Feedback: 1. Non-negative matrix factorization (NMF) plays an important role in learning instance-dependent transition matrix. But except for this paper, NMF and label noise learning are usually not mentioned together. It will be better to provide more related work about NMF or other parts-based learning techniques in main paper or supplementary materials, which will be easier for readers to understand the proposed method. 2. The authors did’t use any clean data in the experiments on synthetic datasets and Clothing1M. I want to know how can this method utilize a small trusted dataset. 3. Some descriptions of experimental settings need to be added. For instance, are data augmentation techniques, e.g., random crop and horizontal flip, used in the experiments? Please give some explanation or emphasis. 4. In this paper, a slack variable is introduced to revise all the instance-dependent transition matrices. I would like to know whether it is possible to introduce different slack variables for different instance-dependent transition matrices. 5. All the datasets used in the experiment are image datasets, i.e., MNIST, Fashion-MNIST, SVHN, CIFAR-10 and Clothing1M. The authors employ NMF to learn parts of image objects and combination parameters. Note that NMF also can be exploited for text data to learn the parts of sentences, i.e., some key words. However, the authors don’t provide experimental results on the text dataset. It will be more convincing if the authors can verify the effectiveness of the proposed methods through experiments on text datasets.


Review 4

Summary and Contributions: The paper is regarding to learning with the instance-dependent label noise. The authors approximated the instance-dependent transition matrix for an instance by a combination of the transition matrices for the parts of the instance. The transition matrices for parts were learned by exploiting data points that belong to a specific class almost surely.

Strengths: The authors proposed an assumption for instance-dependent label noise: The noise depends only on parts of instances. The authors also proposed how to learn parts-dependent transition matrices.

Weaknesses: Descriptions on "the state-of-the-art approaches" for learning from the instance-dependent label noise are not sufficient. Authors stated "Empirical evaluations on synthetic and real-world datasets demonstrate our method is superior to the state-of-the-art approaches for learning from the instance-dependent label noise." What is the rationale of achieving the outperformance theoretically and methodologically?

Correctness: Looks correct.

Clarity: The writing can be improved.

Relation to Prior Work: Can be improved.

Reproducibility: Yes

Additional Feedback: After reading the feedback from the authors, I found that the following question is not sufficiently addressed. "What is the rationale of achieving the outperformance theoretically and methodologically?" There is no theoretical justification for the proposed method in this manuscript. There are lots of existing nonnegative-matrix-factorization related methods and variants. The discussion of "the state-of-the-art approaches" is not sufficient.

[Author Response · NeurIPS 2020]

We thank all reviewers for providing us valuable and insightful comments. Below, we answer all of the questions.

**R1** Q1) The main assumption in this method for part-dependent label noise is not realistic.

A1) There are lots of psychological and physiological evidences showing that we human perceive objects starting
from their parts. Thus, we believe the assumption makes sense in reality. In this paper, we first learn the parts of
instances using all training examples rather than only using anchor points. Then, we learn the parts-dependent transition
matrices by exploiting anchor points. Lastly, the instance-dependent transition matrix can be well approximated by a
combination of the parts-dependent transition matrices. We think the reason for your concern may be resulted by our
description or presentation, and we will make them clearer in our final version.

**R1** Q2) When the deep model is trained using the noisy labels, the features maybe not the accurate or reliable features.
A2) We agree with this point. The memorization effect of deep neural networks shows that they would first memorize
training data of clean labels and then those of noisy labels. Therefore, we can perform early stopping with a noisy/clean
validation set to prevent deep neural networks from overfitting noisy labels. Then, we can obtain relatively reliable
features. This strategy is widely used in existing works such as Forward [44] and T-Revision [58].

**R1** Q3) Please be more specific in references and elaborate on the conjecture of the parameters for combination.
A3) Thanks. We are the first to do so, and we will be more specific in discussions and elaborate the conjecture clearly.

**R1** Q4) It would be nice to visualize parts for some classes of datasets used in the experiments, like Figure 1.
A4) Due to the page limit, we will visualize parts for some classes of datasets in our final supplementary material.

**R2** Q5) The presentation of the subsection "Learning the parts-dependent transition matrices" could be further improved.
A5) We will take your advice and improve the presentation to make the key point clearer to readers.

**R2** Q6) For the experimental results, it would be better to show whether the performance gap is significant or not.
A6) We have taken your advice and conducted significance tests. Almost all results were statistically significant. Due to
the page limit, we will provide detailed experimental results in our final supplementary material.

**R3** Q7) It will be better to provide more related work about NMF or other parts-based learning techniques.
A7) We will add more detailed related work about NMF or other parts-based learning techniques in this paper.

**R3** Q8) How can the proposed method utilize a small trusted dataset in the experiments?
A8) If a small trusted dataset is available, it is helpful. It can be used to (1) better learn the parts/instance-dependent
transition matrix (2) better validate the selected deep learning model, and to (3) fine-tune the deep neural networks.

**R3** Q9) Please give some explanation or emphasis for data augmentation techniques.
A9) We don't use any data augmentation technique. We will emphasize this in the experiment settings.

**R3** Q10) Whether it is possible to introduce different slack variables for instance-dependent transition matrices?
A10) It is possible, but we believe the current way of introducing a single slack variable to revise instance-dependent
transition matrices is better. If we introduce different slack variables for different instance-dependent transition matrices,
this will make optimization difficult as there are too many parameters in slack variables to learn.

**R3** Q11) Please verify the effectiveness of the proposed methods through experiments on text dataset.
A11) We verify our approach on the text dataset *NEWS* (http://qwone.com/~jason/20Newsgroups/). A network
architecture with three convolutional layers and one fully connected layer is employed. Other experiment settings are
the same as those in this paper. We follow Co-teaching+ and borrow the pre-trained word embeddings from GloVe
(https://nlp.stanford.edu/projects/glove/). The results are presented in Table 1. We can see that the proposed method
consistently outperforms the baselines.

Table 1: Means and standard deviations of classification accuracy on *NEWS*.

| | IDN-10% | IDN-20% | IDN-30% | IDN-40% | IDN-50% |
|---|---|---|---|---|---|
| CE | 69.58±0.42 | 66.80±0.36 | 63.11±0.74 | 58.37±0.88 | 54.75±1.62 |
| Decoupling | 69.35±0.41 | 65.32±0.43 | 58.75±0.84 | 51.63±0.77 | 43.05±1.52 |
| MentorNet | 69.03±0.35 | 66.92±0.54 | 62.87±1.31 | 54.35±1.21 | 48.35±1.45 |
| Co-teaching | 69.37±0.29 | 67.99±0.76 | 64.15±0.89 | 56.36±0.71 | 52.32±1.03 |
| Co-teaching+ | 69.35±0.73 | 64.03±0.91 | 56.37±0.61 | 41.88±1.74 | 10.78±5.87 |
| Joint | 69.73±0.51 | 67.45±0.49 | 64.54±0.74 | 60.67±0.83 | 56.72±2.10 |
| DMI | 70.35±0.62 | 68.01±0.45 | 64.28±0.61 | 60.73±0.62 | 56.33±1.35 |
| Forward | 69.24±0.45 | 66.01±0.55 | 62.07±0.58 | 56.33±0.71 | 53.25±1.43 |
| Reweight | 70.25±0.30 | 68.42±0.77 | 65.05±0.93 | 59.37±1.32 | 57.31±3.51 |
| T-Revision | 70.72±0.32 | 69.91±0.49 | 67.28±0.81 | 61.78±0.99 | 59.29±2.07 |
| PTD-F | 70.01±0.47 | 66.78±0.68 | 62.16±0.77 | 59.54±0.63 | 53.63±1.31 |
| PTD-R | 71.03±0.45 | 70.02±0.53 | 68.32±0.72 | 62.37±0.45 | 62.01±1.21 |
| PTD-F-V | 70.27±0.28 | 66.81±0.48 | 62.80±0.76 | 59.71±0.46 | 54.23±1.17 |
| PTD-R-V | **71.92±0.34** | **71.33±0.42** | **69.01±0.85** | **63.17±0.58** | **62.77±0.98** |

**R4** Q12) Descriptions on the state-of-the-art
approaches are not sufficient.

A12) We will add descriptions on the state-
of-the-art approaches in this paper.

**R4** Q13) What is the rationale of achieving
the outperformance theoretically and method-
ologically?

A13) Existing methods focus on class-
dependent noise, but in many real-world ap-
plications, noise can be instance-dependent.
Then the existing methods give a biased solu-
tion. On the other hand, completely instance-
dependent noise is too flexible and is actually not identifiable in general. In this paper, we consider an intermediate and
practical situation of parts-dependent noise, which produces a less biased but still reliable solution. To be specific, there
are many psychological and physiological evidences showing that we human perceive instances by decomposing them
into parts. Motivated by these evidences, we learn the parts of instances and use the combination of parts-dependent
transition matrices to well approximate instance-dependent transition matrix. Empirical results demonstrate that our
proposed method consistently outperforms existing methods.

[Meta-Review · NeurIPS 2020]

Novel and interesting paper that is very relevant to the conference. Experimental study demonstrates the validity of the ideas. Please use the reviewers advise to improve the paper.